# The interaction between rumination and job embeddedness among trainee nurses during internship: A multicenter longitudinal study

**Weiyi Wang[1], Huizhen Ma[1], Lei Huang[2], Chen Ying[2]***

**1** Department of Geriatric Medicine, Nanjing Drum Tower Hospital: The Affiliated Hospital of Medical School, Nanjing University, Nanjing, China, **2** Phase I Clinical Trials Unit, Nanjing Drum Tower Hospital: The Affiliated Hospital of Medical School, Nanjing University, Nanjing, China

◉ These authors contributed equally to this work.
* 13814021690@163.com

## Abstract

### Objective

To explore the developmental trajectory of rumination and job embeddedness among college trainee nurses and examine the predictive relationship between them, with the aim of providing a theoretical basis for improving trainee nurses' job embeddedness.

### Design

A longitudinal study.

### Methods

A total of 405 nursing interns from 12 colleges and universities in China were selected as the survey subjects. They were tracked using the Ruminative Responses Scale (RRS) and the Global Job Embeddedness Items (GJEI) at three time points: T1 (initial stage of internship, 1–2 months), T2 (middle stage of internship, 5–6 months), and T3 (completion of internship, 9–10 months). A cross-lagged model was constructed to analyze the causal relationship between the two variables.

### Results

A total of 382 valid questionnaires were collected, yielding an effective response rate of 94.32%. In analyzing the overall trend from T1 to T3, rumination significantly decreased from a baseline of ($44.79 \pm 5.28$) to ($41.61 \pm 5.27$) ($F = 47.360$, $P < 0.001$). Concurrently, the score of job embeddedness increased significantly from ($22.42 \pm 4.89$) to ($25.48 \pm 4.60$) ($F = 25.690$, $P < 0.001$). The level of rumination at a previous time point negatively predicted job embeddedness at the subsequent point ($\beta = -0.403$, $P < 0.01$; $\beta = -0.411$, $P < 0.001$), and job embeddedness at a previous

**Data availability statement:** All relevant data are within the paper and its Supporting Information files.

**Funding:** The study was supported by Project of China Hospital Reform and Development Research Institute of Nanjing University, Nanjing Drum Tower Hospital and Aid project of Jiangsu Ningai Medical Development & Medical Aid Foundation (NDYGN2024064).

**Competing interests:** The authors have no conflicts of interest to disclose.

time point negatively predicted rumination at the next point on average ($\beta = -0.397$, $P < 0.001$; $\beta = -0.385$, $P < 0.001$). At the initial level, rumination can predict job embeddedness negatively ($\beta = 0.433$, $P = 0.002$), the initial level of rumination can predict the development speed of job embeddedness negatively ($\beta = 0.395$, $P = 0.005$), and the development speed of rumination can positively predict the development speed of job embeddedness ($\beta = 0.450$, $P < 0.001$).

## Conclusion

practice trainee nurses rumination is on the decline, job embeddedness is on the rise. Educators and clinical instructors should assess and monitor rumination levels in trainee nurses and strategically leverage the dynamic relationship between rumination and job embeddedness to enhance the latter during clinical training.

## Introduction

With the rapid development of medical and healthcare services, the importance of nursing work has become increasingly prominent. Nursing has evolved from providing basic disease care to offering comprehensive services that encompass emotional support, health promotion, and environmental maintenance. Consequently, nurses' professional literacy and attitudes have a profound impact on patients' rehabilitation experiences and quality of life [1,2].

Rumination is a psychological concept referring to a cognitive pattern in which individuals repeatedly and passively dwell on negative emotions and the events that trigger them. It is characterized by persistent recall of personal mistakes and failures, accompanied by self-criticism, helplessness, and a perceived loss of control [3]. During clinical practice, trainee nurses are exposed to high-intensity stressors such as life-and-death situations and medical disputes for the first time [4]. The transition from student to clinical nurse demands rapid role adaptation and accountability for patient care. However, limited clinical proficiency often results in hesitation during decision-making, making trainee nurses particularly susceptible to rumination [5].

By amplifying negative emotions, disrupting goal-directed action, and diminishing responsiveness to environmental shifts, rumination intensifies psychopathological states and perpetuates physiological stress, while simultaneously acting as a transdiagnostic liability factor contributing to anxiety, depression, psychosis, insomnia, and impulsivity [6]. Nursing students with a tendency for rumination may directly influence academic procrastination, resulting in a significant decline in cognitive availability and initiative when executing learning tasks [5]. More critically, such cognitive patterns may lead to learned helplessness, causing some trainee nurses to develop avoidance behaviors in the later stages of their internship. This not only reduces their job embeddedness but also increases turnover intention, thereby hindering the development of a robust nursing talent echelon [7,8].

Accordingly, this study hypothesizes that characterizing the longitudinal trajectories of rumination and work embeddedness among intern nurses will facilitate the identification

of critical peak phases of rumination. Furthermore, we postulate that implementing targeted psychological interventions at these pivotal junctures could alleviate negative affect, thereby ultimately enhancing their work embeddedness during the internship period. Building on this premise, we further speculate that dynamically parsing the interactive trajectories of these two constructs will enable the precise identification of intervention 'inflection points.' We hypothesize that interventions administered at these inflection points—aimed at reducing rumination levels and bolstering work embeddedness—may theoretically exert a positive influence on strengthening professional identity and optimizing the quality of nursing services.

Employing a longitudinal design, this study utilized a cross-lagged model and a latent growth model to examine the changing trajectories and reciprocal predictive relationships between ruminative thinking and work embeddedness among nursing interns. The findings provide empirical support for developing targeted intervention strategies, optimizing nursing education and management practices, and ultimately enhancing the professional stability and competence of nursing students.

## Materials and methods

### Study design and participants

This multicenter longitudinal survey employed a multicenter convenience sampling method to recruit 405 intern nurses from 12 higher education institutions (including vocational colleges and undergraduate universities) across China between April 1, 2023, and May 30, 2024.

### Sampling strategy and recruitment procedure

The selection of institutions and participants was conducted in two stages. First, at the institutional level, 12 representative universities and their affiliated hospitals were selected based on geographical distribution (covering Eastern, Central, and Western China) and willingness to cooperate. Subsequently, within each participating institution, participants were recruited via convenience sampling from the pool of eligible intern nurses.

### Ethical approval and informed consent

Prior to data collection, ethical approval was obtained from the Ethics Committee of Nanjing Drum Tower Hospital(ID: 2023-0124-02). Questionnaires were distributed only after receiving approval. All participants were fully informed about the study objectives and procedures via online meetings or offline briefings organized with the assistance of the internship management departments. Baseline questionnaires were distributed through the Wenjuanxing platform (a Chinese online survey tool), and data were collected exclusively from those who provided electronic informed consent.

### Inclusion and exclusion criteria

Inclusion criteria were: (1) nursing students who had completed theoretical coursework and officially entered the clinical internship phase; (2) voluntary participation with informed consent. Exclusion criteria were: (1) absence from clinical practice for more than one month due to sick leave or other reasons; (2) missing more than one follow-up assessment.

The sample size was calculated based on the statistical power requirements for the Latent Growth Model [9,10]. Considering a 20% attrition rate, the minimum sample size required was 250 participants (G*Power 3.1, effect size $f^2 = 0.15$, $\alpha = 0.05$, $1-\beta = 0.8$). Ultimately, 405 trainee nurses were included.

## Methods

### Survey tools

**General information questionnaire.** This self-designed questionnaire collected participants' demographic information, including their highest education level and the classification level of the hospital where they conducted their clinical practice.

### Ruminative responses scale (RRS)

The Chinese version of the RRS, revised by Han Xiu et al.[11] consists of 22 items divided into three dimensions: brooding (5 items), symptom rumination (12 items), and reflection (5 items). Responses are rated on a 4-point Likert scale ranging from 1 ("never") to 4 ("always"), with a total score range of 22–88. Higher scores indicate greater rumination tendency. Evaluation criteria: 22–33 points indicate a low level, 34–66 points indicate an medium level, and 37–88 points indicate a high level. The Chinese version of the RRS demonstrated good reliability, with a Cronbach's α coefficient of 0.88 for the overall scale. Internal consistency for the three subscales was also satisfactory, yielding α coefficients of 0.81 for Brooding, 0.91 for Symptom Rumination, and 0.76 for Reflection. Test-retest reliability over a two-week interval was 0.53(P<0.01), indicating acceptable temporal stability among college student samples. Confirmatory factor analysis (CFA) supported the three-factor structure, revealing a good model fit ($\chi^2$/df<3.0, CFI>0.90, TLI>0.90, RMSEA<0.08), thus confirming the scale's three-dimensional construct validity in the Chinese population. In the current study, the Cronbach's α coefficients for the total scale and the subscales were 0.845, 0.880, and 0.832, respectively.

### Global job embeddedness items (GJEI)

Originally developed by Crossley et al.[12,13] and later translated and adapted by Meihua et al.[12,13], the GJEI consists of 7 items measuring job embeddedness on a single dimension. It uses a 5-point Likert scale ranging from 1 ("strongly disagree") to 5 ("strongly agree"). Items 4 and 6 are reverse scored. Higher scores reflect greater job embeddedness. Evaluation criteria: 7–14 points indicate a low level, 15–21 points indicate a relatively low level, 22–28 points indicate a relatively high level, and 29–35 points indicate a high level. The Chinese version of the scale demonstrated good reliability, with a Cronbach's α coefficient of 0.82 for the overall scale. Factor loadings for all items ranged from 0.58 to 0.79, indicating satisfactory internal homogeneity. Test-retest reliability over a two-week interval was 0.85 (P<0.01), suggesting good temporal stability in measuring work embeddedness across different time points. Confirmatory factor analysis (CFA) supported the unidimensional structure, revealing a good model fit ($\chi^2$/df<3.0, CFI=0.96, TLI=0.95, RMSEA=0.07), thus confirming the scale's construct validity in the Chinese nurse population. In the current study, the Cronbach's α coefficients for the total scale and the subscales were 0.885, 0.812, and 0.844, respectively.

### Data collection methods

Data were collected using a self-developed electronic questionnaire distributed via the WenJuanxing platform (Survey ID: 23045560). The questionnaire included demographic items and the two core scales (RRS and GJEI). Participation was voluntary, and the study strictly adhered to the principle of informed consent. A diachronic tracking design was adopted, with three data collection points: T1: Early stage of internship (1–2 months); T2: Middle stage (5–6 months); T3: End stage (9–10 months). The questionnaire platform was configured with a mandatory response mechanism, ensuring that all questions were completed before submission.

### Statistical methods

Data analysis was conducted using SPSS 26.0 and Mplus 8.0. Categorical variables were presented as frequencies and percentages, while continuous variables were expressed as mean±standard deviation. Pearson correlation analysis was used to examine associations between variables. To explore the linear developmental trajectory of rumination and job embeddedness, an unconditional linear LGM was used. The intercept represented the initial level, and the slope reflected the rate of change. A parallel growth model was employed to analyze the dynamic relationship between rumination and job embeddedness. Additionally, a CLM was used to assess the bidirectional predictive relationship between the two variables. Model fit was evaluated using the following indices: Chi-square/degrees of freedom ratio ($\chi^2$/df); Comparative Fit Index (CFI); Tucker–Lewis Index (TLI); Root Mean Square Error of Approximation (RMSEA); Standardized Root Mean

Square Residual (SRMR). Good model fit was indicated by $\chi^2 df < 5.000$, CFI > 0.900, TLI > 0.900, RMSEA < 0.080, and SRMR < 0.100. A p-value of < 0.05 was considered statistically significant.

## Results

### General demographic data

A total of 382 valid questionnaires were collected, yielding an effective response rate of 94.32%. The respondents were college trainee nurses aged 19–27 years, including 43 males (11.26%) and 339 females (88.74%). Detailed demographic information is presented in Table 1.

### Common method deviation test

There were 12, 11, and 10 factors with eigenvalues greater than 1 in the T1, T2, and T3 tests, respectively. The explained variance for the three measurements was 15.18%, 18.55%, and 22.37%, all of which were below the critical threshold of 40% [14].

### The trend of change in rumination and job embeddedness at three time points

As for the overall trend (T1-T3), rumination decreased significantly from ($44.79 \pm 5.28$) at baseline to ($41.61 \pm 5.27$) ($F = 47.360$, $P < 0.001$). The score of job embeddedness increased significantly from ($22.42 \pm 4.89$) to ($25.48 \pm 4.60$) ($F = 25.690$, $P < 0.001$), as shown in Table 2, Figs 1 and 2.

### Correlation analysis of rumination and job embeddedness scores at three time points

Pearson correlation analysis was utilized to examine the relationship between rumination and job embeddedness at three distinct time points. The results indicated a significant correlation between the two variables at all three time points ($P < 0.05$), fulfilling the prerequisites for both the cross-lag model and the latent variable growth model. The matrix representation of these relationships is presented in Table 3.

### Cross-lagged model of rumination and job embeddedness in trainee nurses

Four path models (M1-M4) were constructed to examine the causal relationship between rumination and job embeddedness: (1) M1 (baseline model) featured an autoregressive path (T1→T2→T3) that included only rumination and job embeddedness. M2 (one-way prediction model) built upon M1 by adding the prediction path of rumination on job embeddedness at the subsequent time point (T1_rumination→T2_job embeddedness, T2_rumination→T3_job embeddedness); M3 (reverse prediction model) also expanded upon M1 by incorporating the prediction path of job embeddedness influencing rumination at the next time point (T1_job embeddedness→T2_rumination, T2_job embeddedness→T3_rumination);

Table 1. General information of the respondents ($n = 382$).

| Items | Categories | n | % | Items | Categories | n | % |
|---|---|---|---|---|---|---|---|
| Age (years) | < 20 | 45 | 11.78 | Level of practice hospital | Grade II and below | 25 | 6.54 |
| | 20~25 | 295 | 77.23 | | 3rd Grade B | 56 | 14.66 |
| | > 25 | 42 | 10.99 | | 3rd Grade A | 301 | 78.80 |
| Gender | Male | 43 | 11.26 | Whether he is an only child | Yes | 96 | 25.13 |
| | Female | 339 | 88.74 | | No | 286 | 74.87 |
| Highest degree | College or below | 95 | 24.87 | Place of residence | City | 100 | 26.18 |
| | Undergraduate | 245 | 64.14 | | Township | 153 | 40.05 |
| | Master's degree or above | 42 | 10.99 | | Rural | 129 | 33.77 |

**Table 2. Dynamic changes of rumination and job embeddedness of trainee nurses during internship (*n* = 382).**

| Variables | T1 | T2 | T3 | F/χ2 | P |
|---|---|---|---|---|---|
| **Rumination** | | | | | |
| Total score ($\bar{x} \pm s$) | 44.79±5.28 | 43.47±6.03 | 41.61±5.27 | 47.360 | < 0.001 |
| Percentage of grading [*n* (%)] | | | | 26.903 | < 0.001 |
| Low (22–33 points) | 52 (13.61) | 84 (21.99) | 101 (26.44) | | |
| Medium (34–66 points) | 246 (64.40) | 237 (61.04) | 234 (61.26) | | |
| High (37–88 points) | 84 (21.99) | 61 (15.97) | 47 (12.30) | | |
| **Job embeddings** | | | | | |
| Total score ($\bar{x} \pm s$) | 22.42±4.89 | 23.73±4.80 | 25.48±4.60 | 25.690 | < 0.001 |
| Percentage of grading [*n* (%)] | | | | 34.488 | < 0.001 |
| Low (7–14 points) | 75 (19.63) | 60 (15.71) | 49 (12.83) | | |
| Relatively low (15–21 points) | 135 (35.34) | 101 (26.44) | 92 (24.08) | | |
| Relatively high (22–28 points) | 151 (39.53) | 174 (45.55) | 185 (48.43) | | |
| High (29–35 points) | 21 (5.50) | 47 (12.30) | 56 (14.66) | | |

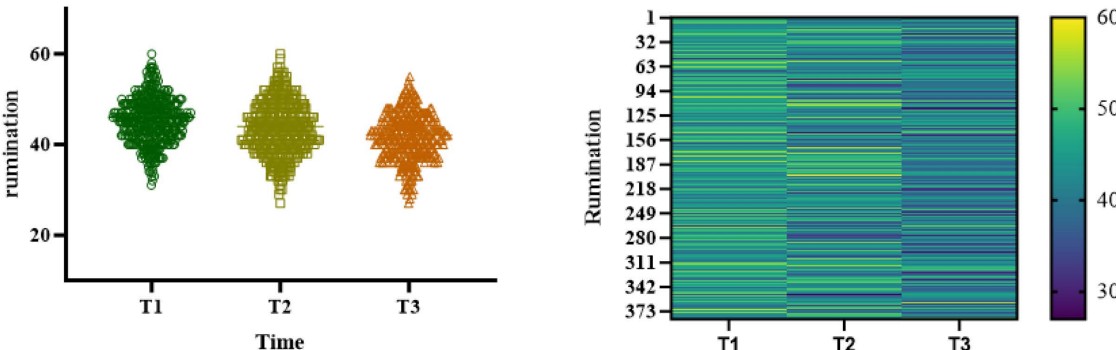

**Fig 1. Heat map distribution and trend map of rumination of trainee nurses during internship.**

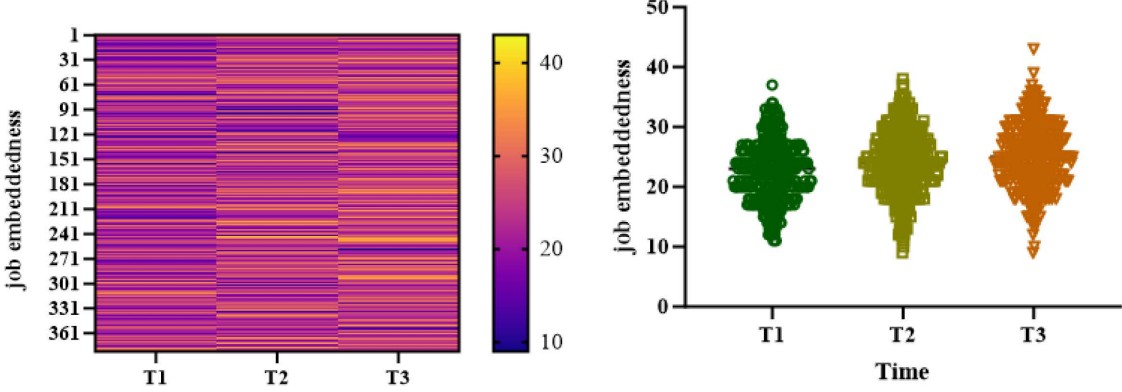

**Fig 2. Heat map distribution and trend of job embeddedness heatmap of trainee nurses during internship.**

**Table 3. Correlation coefficient matrix of rumination and job embeddedness of trainee nurses at three time points (*r*, *n* = 382).**

| Items | ① | ② | ③ | ④ | ⑤ | ⑥ |
|---|---|---|---|---|---|---|
| ① Rumination T1 | 1 | | | | | |
| ② Job embeddedness T1 | −0.439** | 1 | | | | |
| ③ Rumination T2 | 0.542** | −0.337* | 1 | | | |
| ④ Work embedded in T2 | −0.401** | 0.482** | −0.472** | 1 | | |
| ⑤ Rumination T3 | 0.462** | −0.305* | 0.514** | −0.275* | 1 | |
| ⑥ Work embedding T3 | −0.303* | 0.370** | −0.347** | 0.493** | −0.434*** | 1 |

Note:* denotes P<0.05, **denotes P<0.01, and *** P<0.001.

M4 (two-way full model) encompassed both bidirectional paths of rumination→job embeddedness and job embeddedness→rumination. Robust maximum likelihood estimation (MLR) was utilized, and model fit comparisons are detailed in Table 4. Notably, M4 significantly outperformed M1, M2, and M3 ($\Delta\chi^2$ = 137.228, P<0.001), and exhibited the best fit (CFI > 0.984, RMSEA<0.05), lending support to the hypothesis of their interaction.

A CLM was constructed to examine the reciprocal predictive relationship between rumination and job embeddedness. The model demonstrated a good fit: $\chi^2$/df = 2.034, CFI = 0.986, TLI = 0.972, RMSEA = 0.000, and SRMR = 0.037. As shown in Fig 1, rumination at the previous time point significantly and negatively predicted job embeddedness at the subsequent time point ($\beta$ = −0.403, P<0.01; $\beta$ = −0.411, P<0.001). Similarly, job embeddedness at the prior time point significantly and negatively predicted rumination at the next time point ($\beta$ = −0.397, P<0.001; $\beta$ = −0.385, P<0.001). The detailed path relationships are illustrated in Fig 3.

### Trainee nurses' rumination and job embeddedness in parallel latent variables

A parallel latent variable growth model was constructed to examine the longitudinal relationship between rumination and job embeddedness among trainee nurses. The model demonstrated a good fit: $\chi^2$/df = 2.062, CFI = 0.984, TLI = 0.972, RMSEA = 0.048, and SRMR = 0.030. At the initial level, rumination significantly and negatively predicted job embeddedness ($\beta$ = −0.433, P = 0.002), indicating that trainee nurses with higher initial levels of rumination exhibited lower initial levels of job embeddedness. Additionally, the initial level of rumination negatively predicted the growth rate of job embeddedness ($\beta$ = −0.395, P = 0.005), suggesting that higher baseline rumination was associated with a slower increase in job embeddedness over time. Furthermore, the rate of change in rumination positively predicted the rate of change in job embeddedness ($\beta$ = 0.450, P<0.001), meaning that a faster decline in rumination was associated with a more rapid increase in job embeddedness. The specific path relationships are illustrated in Fig 4.

## Discussion

This longitudinal study found that among college trainee nurses, rumination decreased while job embeddedness increased during clinical practice. The clinical practice environment plays a crucial role in shaping these changes.

**Table 4. Model fitting comparison.**

| Model | $\chi^2$/df | CFI | TLI | RMSEA | SRMR | $\Delta\chi^2$ (vs M1) | $\Delta$ df | P |
|---|---|---|---|---|---|---|---|---|
| M1 | 2.350 | 0.932 | 0.914 | 0.078 | 0.069 | | | |
| M2 | 1.814 | 0.960 | 0.942 | 0.064 | 0.048 | 125.123 | 2 | <0.001 |
| M3 | 2.133 | 0.955 | 0.934 | 0.067 | 0.054 | 20.560 | 2 | <0.001 |
| M4 | 1.417 | 0.984 | 0.978 | 0.040 | 0.034 | 137.228 | 4 | <0.001 |

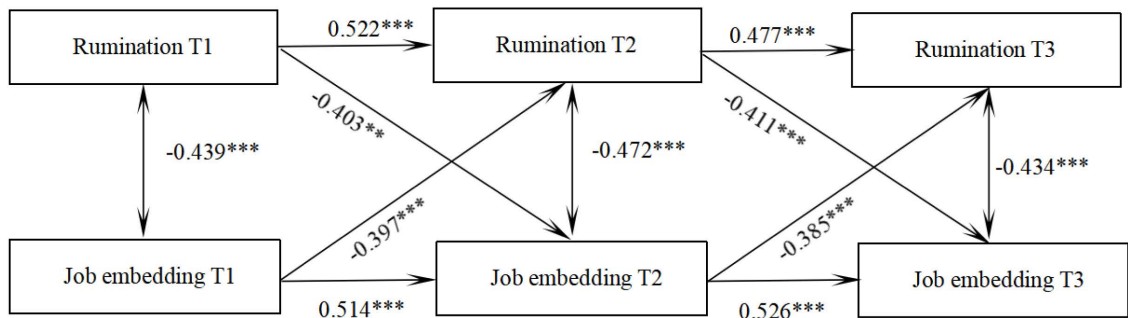

**Fig 3. Predictive pathways of rumination and job embedding of trainee nurses at three time points.**

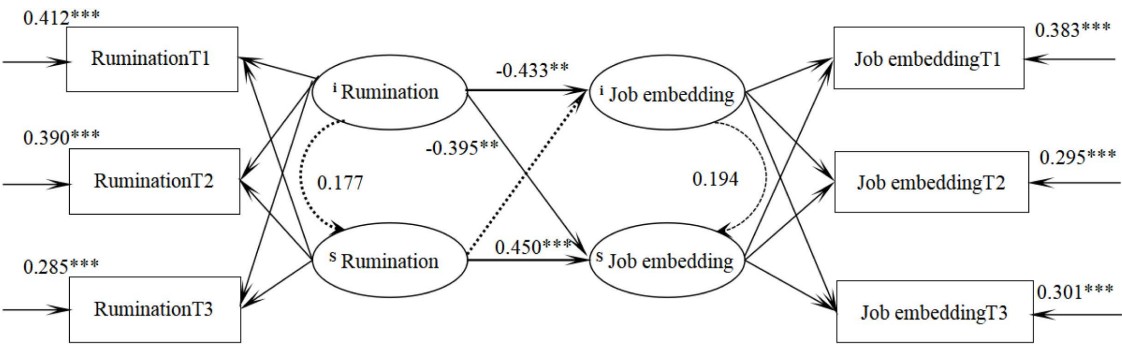

**Fig 4. Parallel latent variable model of Rumination and job embeddedness for trainee nurses.**

Moscaritolo et al. [15] reported that supportive clinical guidance and a positive team atmosphere significantly enhance trainee nurses' self-confidence and sense of belonging, reducing negative emotions during work. The professional qualities and enthusiasm of experienced nurses and clinical instructors subtly influence trainee nurses [16]. Through observation and practice, trainee nurses internalize these behaviors, developing positive professional norms and attitudes [17]. With the progression of clinical practice, nursing students accumulate a sense of professional benefit by successfully resolving clinical problems, receiving positive feedback from patients, and gaining supportive guidance from clinical instructors. This includes perceiving personal growth, a sense of team belonging, and the positive significance of nurse-patient relationships. This deep recognition of the nursing profession's value effectively enhances their self-efficacy [18], thereby redirecting cognitive resources away from immersion in negative emotions towards improving professional skills and strengthening professional identity. Consequently, this process helps mitigate the vicious cycle of ruminative thinking. Consequently, rumination declines and job embeddedness rises over the internship period. The internship offers trainee nurses opportunities to engage in nursing work and experience the fulfillment of helping Participants. Moreover, the guidance and role modeling by clinical teachers deepen their understanding of the nursing profession's value [19]. Caring for Participants and addressing nursing challenges further strengthens their professional identity and sense of mission, which clarifies their understanding of nursing and reinforces their commitment to the profession, reflected in increasing job embeddedness scores [20].

Employing a CLM, this longitudinal study revealed significant reciprocal relationships between ruminative thinking and work embeddedness among nursing interns during their clinical practice. Specifically, higher levels of ruminative thinking at one time point significantly predicted lower work embeddedness at the subsequent time point, while conversely, greater

work embeddedness at one time point significantly predicted lower ruminative thinking at the next. This indicates a mutual influence between the two variables, suggesting the potential for a self-reinforcing cycle over time.

As a passive, repetitive negative cognitive pattern, ruminative thinking consumes substantial cognitive resources. This depletion leaves fewer mental resources for clinical focus, complex problem-solving, and active integration into the team, thereby directly hindering the development of deeper work embeddedness. Conversely, a high level of work embeddedness, characterized by strong organizational ties, a sense of fit, and perceived value, fosters positive emotional states and enhances emotion regulation capabilities. This supportive state helps individuals disengage more quickly from negative events, thereby reducing the frequency and intensity of ruminative thinking [21]. This improvement in emotion regulation efficiency enables interns to cope with clinical pressures with greater stability, effectively disrupting the vicious cycle of rumination. The cumulative nature of this interaction over time is clearly captured by the cross-lagged predictive effects observed in the model. From the perspective of cognitive resource competition, rumination fosters indulgence in negative emotions, hindering professional role cognition and weakening organizational commitment weakening [22]. Work embeddedness may reinforce professional identity, thereby channeling cognitive resources toward professional engagement and potentially buffering against the onset of rumination. Concurrently, from a theoretical standpoint, the positive emotional experiences engendered by high work embeddedness hold promise for disrupting the negative cycle of rumination.

The parallel latent variable growth model showed that at the initial level, rumination negatively predicted job embeddedness; in other words, the higher the initial rumination of trainee nurses, the lower their initial job embeddedness. High rumination accelerates the loss of professional enthusiasm due to ongoing emotional exhaustion, which hinders cognitive reconstruction of professional roles, causing doubts about nursing work and reducing job embeddedness. Additionally, the initial level of rumination negatively predicted the development rate of job embeddedness, higher initial rumination corresponded to a slower increase in job embeddedness. Trainee nurses with elevated rumination tend to engage in negative thinking, such as repeatedly recalling operational errors, which forms a rigid cognitive pattern of rumination and self-denial. This leads to a continuous decline in self-efficacy and reduced willingness to engage in work. Excessive rumination also consumes cognitive resources, making it harder to manage clinical challenges effectively and impeding the development of professional competence. Moreover, rumination-induced negative emotions lower sensitivity to organizational support, reduce proactive seeking of mentor or peer guidance, and delay the formation of professional identity and organizational commitment, thus slowing job embeddedness growth. Conversely, the development speed of rumination positively predicted the development speed of job embeddedness: the faster rumination decreased, the faster job embeddedness increased. When the rate of decline in ruminative thinking among nursing interns accelerates, their persistent immersion in negative thoughts is alleviated, freeing up cognitive resources that can be redirected toward professional learning and skill enhancement. This contributes to an increase in their professional self-efficacy. Simultaneously, improved efficiency in emotion regulation enables them to engage in clinical work with a more positive and stable mindset, which may further enhance their level of vocational engagement and positively influence the deepening of work embeddedness [23].

Based on the findings, this study offers significant implications for clinical nursing education management. First, it is essential to establish an early screening and tiered intervention mechanism to systematically assess nursing students' levels of ruminative thinking at the beginning of their internships. Individuals with high ruminative tendencies should receive preemptive psychological support, focusing on redirecting negative, obsessive thoughts toward constructive reflection, thereby mitigating initial barriers to work embeddedness. Second, clinical teaching should deepen cognitive restructuring training by integrating strategies such as mindfulness meditation and cognitive-behavioral therapy into daily mentoring. This approach helps enhance students' awareness of negative emotions and fosters non-judgmental acceptance, effectively reducing the duration and intensity of rumination and reallocating cognitive resources toward professional identity formation.

Furthermore, there is an urgent need to build a multi-level supportive clinical environment by strengthening preceptors' coaching leadership skills. Cultivating positive mentor-mentee relationships and team dynamics can enhance students'

sense of organizational belonging, person-job fit, and perceived cost of leaving, thereby systematically elevating their work embeddedness. In addition, institutional interventions should emphasize the regular reinforcement of positive professional experiences. Examples include structured documentation and reflection on successful nursing cases, establishing patient feedback mechanisms, and organizing sharing sessions on perceived professional benefits. These measures help consolidate and boost nursing students' self-efficacy and sense of professional value.

Finally, administrators may adopt a cyclical evaluation and dynamic feedback model to regularly monitor the interaction between ruminative thinking and work embeddedness. When a significant decline in rumination is detected, offering more challenging clinical tasks and career development opportunities can maximize the "cognitive liberation" window, accelerating the growth of work embeddedness. In summary, by creating a supportive environment, cultivating proactive cognitive strategies among students, and implementing dynamic assessments and interventions, it is possible to effectively break the vicious cycle of high rumination and low work embeddedness, thereby fostering the positive professional psychological development of nursing students.

## Conclusions

Rumination among trainee nurses showed a downward trend, while job embeddedness showed an upward trend. The initial level of rumination negatively predicted both the initial level and the development speed of job embeddedness. Additionally, the development speed of rumination positively predicted the development speed of job embeddedness. College and clinical instructors should monitor the rumination levels of trainee nurses and strategically leverage the interaction between rumination and job embeddedness to enhance job embeddedness and strengthen professional identity. This study's longitudinal design tracked changes in rumination and job embeddedness among 382 trainee nurses throughout their internship, providing valuable insights for the clinical understanding of their development.

However, some limitations exist. Despite being a multi – center study, the relatively small sample size may reduce statistical power. Moreover, the use of self – reported measures in the survey may increase the risk of bias.

Future studies could incorporate objective assessment tools to further elucidate the dynamic interaction mechanisms between the two variables.

Lastly, while the study provides evidence of dynamic associations between variables, it cannot establish strict causal relationships as achieved in randomized controlled trials. There are also possible attrition bias across the three time points, limited generalizability beyond the selected Chinese colleges, and a lack of assessment of external factors (e.g., organizational support, clinical workload) that might influence both rumination and job embeddedness.

## Supporting information

**S1 Data. Data.**
(XLSX)

## Author contributions

**Conceptualization:** Weiyi Wang, Huizhen Ma, Lei Huang, Chen Ying.

**Data curation:** Weiyi Wang, Chen Ying.

**Formal analysis:** Huizhen Ma.

**Funding acquisition:** Chen Ying.

**Investigation:** Weiyi Wang, Huizhen Ma, Lei Huang, Chen Ying.

**Resources:** Chen Ying.

**Supervision:** Weiyi Wang.

**Visualization:** Weiyi Wang.

**Writing – original draft:** Weiyi Wang, Huizhen Ma, Lei Huang, Chen Ying.

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
