## [Decision Letter · Decision Letter 0]

5 Jan 2026

PONE-D-25-60904The interaction between rumination and job embeddedness among trainee nurses duringinternship: a multicenter longitudinal studyPLOS One

Dear Dr. wang,

Thank you for submitting your manuscript to PLOS ONE. After careful consideration, we feel that it has merit but does not fully meet PLOS ONE’s publication criteria as it currently stands. Therefore, we invite you to submit a revised version of the manuscript that addresses the points raised during the review process.

Please submit your revised manuscript by Feb 19 2026 11:59PM. If you will need more time than this to complete your revisions, please reply to this message or contact the journal office at plosone@plos.org. Please include the following items when submitting your revised manuscript:

We look forward to receiving your revised manuscript.

Kind regards,

Ahmed Abdelwahab Ibrahim El-Sayed

Academic Editor

PLOS One

Journal Requirements:

“The study was supported by Project of China Hospital Reform and Development Research Institute of Nanjing University ,Nanjing Drum Tower Hospital and Aid project of Jiangsu Ningai Medical Development & Medical Aid Foundation(NDYGN2024064).”

3. Thank you for providing your underlying data as Supporting Information.

We note that the data set contains text or data that is not in English. Please note that PLOS is an English-language publisher, so we require data sets to be provided in English as well. Please upload an English-language version of your data set.

This will also allow us to determine if your data follows PLOS standards per our Data Availability policy here: https://journals.plos.org/plosone/s/data-availability

**Additional Editor Comments:**

Dear Authors,

Thank you for submitting your manuscript to PLOS ONE. Based on the reviewers’ evaluations, several substantive concerns have been raised that must be thoroughly addressed before further consideration of your manuscript.

We encourage you to carefully review the comments and submit a revised version that responds to each point.

Reviewers' comments:

Reviewer's Responses to Questions

**Comments to the Author**

1. Is the manuscript technically sound, and do the data support the conclusions?

Reviewer #1: Partly

Reviewer #2: Yes

Reviewer #3: Yes

Reviewer #4: Yes

2. Has the statistical analysis been performed appropriately and rigorously? 

Reviewer #1: Yes

Reviewer #2: Yes

Reviewer #3: Yes

Reviewer #4: Yes

3. Have the authors made all data underlying the findings in their manuscript fully available?

Reviewer #1: Yes

Reviewer #2: Yes

Reviewer #3: Yes

Reviewer #4: No

4. Is the manuscript presented in an intelligible fashion and written in standard English?

Reviewer #1: Yes

Reviewer #2: Yes

Reviewer #3: Yes

Reviewer #4: No

5. Review Comments to the Author

Reviewer #1: Title: The interaction between rumination and job embeddedness among trainee nurses

duringinternship: a multicenter longitudinal study.

I. Overall

II. Abstract

1. Although this is underpinned by the three key moments you described later, I think that clearly mentioning the study design is a non-negotiable requirement.

2. Please specify the instruments used in the abstract.

3. Keywords: Please, use the exact MeSH terms for the "Keywords" section to ensure better indexing.

III. Introduction

1. Line 44 ► What do you mean by “participants”?

2. Lines 48-50 ►“During clinical practice […] first time [4]”: I think that reference [4] does not support this claim.

3. Lines 50-53 ►“The transition […] practice stage [5]”: I think that reference [5] does not support this claim.

4. Lines 54-56 ►“Research has shown […] attentional resources [6]”: I think that reference [6] does not claim.

5. Lines 58-62 ►“More critically […] nursing talent echelon [8, 9]”: I think that reference [8,9] does NOT support the claim you provided, because both references refer to patients, not nursing students.

6. Lines 63-65 ►“Understanding the developmental trajectory […] job embeddedness [10]”: Since the reference [10] never measured job embeddedness, it cannot statistically support a claim about enhancing it.

7. Please revise the introduction to ensure every factual claim and contextual statement is supported by a robust, peer-reviewed source. This is critical for academic rigor and for accurately framing the research gap your work aims to address.

8. Lines 72-74 ► Lines 63-65 ► I think that your study is an empirical study, the phrase "aims to provide a theoretical foundation" might be slightly overstating the primary goal.

IV. Materials and methods

1. In adherence to the journal’s submission guidelines, please revise the section heading ‘Research methods’ to ‘Materials and Methods’.

2. Reorganize your Methods section with clear, non-overlapping subheadings.

3. The claim that "at least 200 cases were needed" for the LGM must be supported.

4. For an LGM, a simulation-based power analysis is the gold standard. Please provide details: target parameter, assumed effect size (with justification from prior literature), number of waves, and software/method used. Without this, the sample size is arbitrary.

5. Please provide a citation or empirical rationale for the anticipated 20% attrition rate.

V. Results

1. Table 2 shows an inconsistency in the percentage of rating categories.

2. What is your reference for establishing the rating categories for the two scales? Normally, these categories should be defined in the methods.

3. Reference [13] is included in the list of references, but is not cited in the text.

4. Figure 4 requires a legend to facilitate its reading.

VI. Discussion

1. Lines 207-209 ► “Bhuiya et al. [15] […] during work”: This claim does not belong to the reference [15]. The latter is about medical students, not nurses.

2. Lines 212-215 ►“Consequently […] nursing profession’s value [18,19 ]: It appears that reference [18] is entirely fictitious.

3. Lines 216-218 ►“Caring for Participants […] job embeddedness scores”: This claim need to be supported by at least one reference.

4. Lines 221-223 ►“From the perspective […] commitment weakening [5]”: This claim is not supported by the reference [5]. Actually, the key finding of the latter is that rumination is not only a direct pathway to depression but also a trigger for social withdrawal, which further deepens depressive symptoms.

5. Lines 225-226 ►“Emotionally […] negative cycle of rumination [20, 21]”: The two cited references do not provide support for the concept of "job embeddedness," as they do not investigate this construct. To support this claim, please cite literature that specifically defines and examines job embeddedness.

6. Lines 234-247 ►“Trainee nurses […] job embeddedness process [22]”: This 15-line discussion passage contains overinterpretation, distortion of facts, or speculation presented as evidence.

7. Overall, the discussion section lacks synthesis and scholarly depth. I strongly advise you to review the discussion section in question and ensure every sentence that engages with a source can be directly and justifiably traced back to that source’s actual content.

VII. References

1. “Meihua et al.” is cited in the text but absent from the reference list.

2. “Crossley et al. ” is formatted incorrectly in the text per the journal style.

3. What does the symbol [J] at the end of the titles mean?

4. It would be wise to include the DOI when possible.

5. Please verify the title of the reference [3].

6. The entries for these references are incomplete (eg. [11], [14], [17]).

7. It appears that reference [18] is entirely fictitious.

Reviewer #2: The revised manuscript entitled “The interaction between rumination and job embeddedness among trainee nurses during internship: a multicenter longitudinal study” is generally well structured and clearly written. The introduction is informative; however, the authors should carefully verify that the references cited are fully consistent with the statements made. The methodology is clearly described, although the sampling method should be explicitly stated. The results are well presented and appropriately discussed. In addition, the limitations section should be expanded to include aspects beyond the small sample size. Finally, the authors are encouraged to carefully check and standardize all references according to the Vancouver referencing style.

Detailed comments and specific remarks are provided in the attached file.

Reviewer #3: O manuscrito é tecnicamente sólido por apresentar um delineamento longitudinal adequado, com a coleta de dados realizada em três momentos, com uso de abordagens estatísticas robustas e confiáveis (Modelo de defazagem cruzada e modelo paralelo de crescimento latente). A descrição dos métodos foi realizada de maneira clara e apropriada para as respostas as questões da pesquisa. As análises estatísticas forão rigorosas e apresentadas de forma bem transparente. Os indices de ajuste dos modelos utilizados forão reportados adequadamente, e o teste de viés reforçou a confiabilidade dos resultados.

É uma pesquisa original e de relevancia por analisar a ruminação e o enraizamento no trabalho em enfermeiros estagiários, devido principalmente ao contexto de formação clínica.

As conclusões vão de encontro aos dados apresentados , todas as afirmações são sustentadas. O manuscrito é claro, bem estruturado e intelegível para todos, as tabelas são pertinentes e contribuem para compreensão dos achados.

Foi apresentada a aprovação do Comitê de Ética e Pesquisa, e o estudo informou a disponibilidade dos dados de acordo com as diretrizes da revista.

Algumas considerações: Sugiro embasar mais a discussão articulando com modelos teóricos clássicos e contempoâneos sobre o estresse ocupacional, adaptação profissional e saúde mental dos profissionais de saúde. Falar um pouco mais sobre as características culturais e organizacionais do contexto chinês, acrescentar exemplos específicos de intervenções institucionais aplicaveis ao programa de estágio em enfermagem.

Realizar pequenos ajustes na linguagem para melhorar a fluidez, em trechos maiores. Garantir que todas as siglas e abreviações estatísticas sejam definidas na primeira menção no texto.

Reviewer #4: The study presents a sound longitudinal design and an appropriate use of advanced statistical models; however, its contribution to disciplinary knowledge in nursing is limited. The inverse relationship between rumination and job embeddedness has already been widely documented in the literature, particularly in contexts where employability expectations influence the psychological wellbeing of nurses in training. The findings largely confirm predictable trends without offering new conceptual frameworks, additional explanatory variables, or innovative practical implications for nursing education or management. Furthermore, the construct of rumination is not sufficiently contextualised from a nursing perspective. Overall, the study stands out more for its methodological rigour than for making a substantive contribution to the advancement of nursing knowledge.

6. PLOS authors have the option to publish the peer review history of their article (what does this mean?). If published, this will include your full peer review and any attached files.

Reviewer #1: No

Reviewer #2: **Yes:**EL FADELY ABDELMONIM

Reviewer #3: No

Reviewer #4: No

---

## [Author Response · Author response to Decision Letter 1]

29 Jan 2026

Dear Editors and Reviewers,

On behalf of our research team, I would like to express our sincere gratitude for your valuable time and insightful comments on our manuscript. We have carefully addressed each point raised in the review and have revised the manuscript accordingly. We sincerely hope that our responses and revisions meet with your approval.

Yours sincerely,

Weiyi Wang

View Letter

Date: Jan 05 2026 01:08AM

To: "weiyi wang" anotherweiyi@163.com

From: "PLOS ONE" plosone@plos.org

Subject: PLOS ONE Decision: Revision required [PONE-D-25-60904]

Attachment(s):  PONE-D-25-60904 (R).pdf

PONE-D-25-60904

The interaction between rumination and job embeddedness among trainee nurses duringinternship: a multicenter longitudinal study

PLOS One

Dear Dr. wang,

Thank you for submitting your manuscript to PLOS ONE. After careful consideration, we feel that it has merit but does not fully meet PLOS ONE’s publication criteria as it currently stands. Therefore, we invite you to submit a revised version of the manuscript that addresses the points raised during the review process.

Please submit your revised manuscript by Feb 19 2026 11:59PM. If you will need more time than this to complete your revisions, please reply to this message or contact the journal office at plosone@plos.org. Please include the following items when submitting your revised manuscript:

A letter that responds to each point raised by the academic editor and reviewer(s). You should upload this letter as a separate file labeled 'Response to Reviewers'.

We look forward to receiving your revised manuscript.

Kind regards,

Ahmed Abdelwahab Ibrahim El-Sayed

Academic Editor

PLOS One

Journal Requirements:

“The study was supported by Project of China Hospital Reform and Development Research Institute of Nanjing University ,Nanjing Drum Tower Hospital and Aid project of Jiangsu Ningai Medical Development & Medical Aid Foundation(NDYGN2024064).”

Response: This point has been added to the manuscript.

3. Thank you for providing your underlying data as Supporting Information.

We note that the data set contains text or data that is not in English. Please note that PLOS is an English-language publisher, so we require data sets to be provided in English as well. Please upload an English-language version of your data set.

This will also allow us to determine if your data follows PLOS standards per our Data Availability policy here: https://journals.plos.org/plosone/s/data-availability

Response: Data has been updated.

Response: This point has been addressed in the manuscript.

Additional Editor Comments:

Dear Authors,

Thank you for submitting your manuscript to PLOS ONE. Based on the reviewers’ evaluations, several substantive concerns have been raised that must be thoroughly addressed before further consideration of your manuscript.

We encourage you to carefully review the comments and submit a revised version that responds to each point.

Reviewers' comments:

Reviewer's Responses to Questions

Comments to the Author

1. Is the manuscript technically sound, and do the data support the conclusions?

Reviewer #1: Partly

Reviewer #2: Yes

Reviewer #3: Yes

Reviewer #4: Yes

2. Has the statistical analysis been performed appropriately and rigorously?

Reviewer #1: Yes

Reviewer #2: Yes

Reviewer #3: Yes

Reviewer #4: Yes

3. Have the authors made all data underlying the findings in their manuscript fully available?

Reviewer #1: Yes

Reviewer #2: Yes

Reviewer #3: Yes

Reviewer #4: No

4. Is the manuscript presented in an intelligible fashion and written in standard English?

Reviewer #1: Yes

Reviewer #2: Yes

Reviewer #3: Yes

Reviewer #4: No

5. Review Comments to the Author

Reviewer #1: Title: The interaction between rumination and job embeddedness among trainee nurses during internship: a multicenter longitudinal study.

I. Overall

II. Abstract

1. Although this is underpinned by the three key moments you described later, I think that clearly mentioning the study design is a non-negotiable requirement.

Response: The supplementary content has been added to the abstract section.

2. Please specify the instruments used in the abstract.

Response: The necessary revisions have been made to the abstract.

3. Keywords: Please, use the exact MeSH terms for the "Keywords" section to ensure better indexing.

Response: The revisions have been made as requested.

III. Introduction

1. Line 44 ► What do you mean by “participants”?

Response: The term has been changed to 'patients'.

2. Lines 48-50 ►“During clinical practice […] first time [4]”: I think that reference [4] does not support this claim.

3. Lines 50-53 ►“The transition […] practice stage [5]”: I think that reference [5] does not support this claim.

4. Lines 54-56 ►“Research has shown […] attentional resources [6]”: I think that reference [6] does not claim.

5. Lines 58-62 ►“More critically […] nursing talent echelon [8, 9]”: I think that reference [8,9] does NOT support the claim you provided, because both references refer to patients, not nursing students.

6. Lines 63-65 ►“Understanding the developmental trajectory […] job embeddedness [10]”: Since the reference [10] never measured job embeddedness, it cannot statistically support a claim about enhancing it.

7. Please revise the introduction to ensure every factual claim and contextual statement is supported by a robust, peer-reviewed source. This is critical for academic rigor and for accurately framing the research gap your work aims to address.

Response: Thank you for your feedback. We have verified and revised the relevant references accordingly.

8. Lines 72-74 ► Lines 63-65 ► I think that your study is an empirical study, the phrase "aims to provide a theoretical foundation" might be slightly overstating the primary goal.

Response: The relevant wording has been revised.

IV. Materials and methods

1. In adherence to the journal’s submission guidelines, please revise the section heading ‘Research methods’ to ‘Materials and Methods’.

Response: The revisions have been made as requested.

2. Reorganize your Methods section with clear, non-overlapping subheadings.

Response: The revisions have been made as requested.

3. The claim that "at least 200 cases were needed" for the LGM must be supported.

Response: The references have been added.

4. For an LGM, a simulation-based power analysis is the gold standard. Please provide details: target parameter, assumed effect size (with justification from prior literature), number of waves, and software/method used. Without this, the sample size is arbitrary.

Response: The revisions have been made as requested.

5. Please provide a citation or empirical rationale for the anticipated 20% attrition rate.

Response: The revisions have been made as requested.

V. Results

1. Table 2 shows an inconsistency in the percentage of rating categories.

Response: The data has been verified.

2. What is your reference for establishing the rating categories for the two scales? Normally, these categories should be defined in the methods.

Response: The evaluation criteria have been added at the introduction section of the scale.

3. Reference [13] is included in the list of references, but is not cited in the text.

Response: The full list of references has been verified.

4. Figure 4 requires a legend to facilitate its reading.

Response: It has been supplemented in the article.

VI. Discussion

1. Lines 207-209 ► “Bhuiya et al. [15] […] during work”: This claim does not belong to the reference [15]. The latter is about medical students, not nurses.

2. Lines 212-215 ►“Consequently […] nursing profession’s value [18,19 ]: It appears that reference [18] is entirely fictitious.

3. Lines 216-218 ►“Caring for Participants […] job embeddedness scores”: This claim need to be supported by at least one reference.

4. Lines 221-223 ►“From the perspective […] commitment weakening [5]”: This claim is not supported by the reference [5]. Actually, the key finding of the latter is that rumination is not only a direct pathway to depression but also a trigger for social withdrawal, which further deepens depressive symptoms.

5. Lines 225-226 ►“Emotionally […] negative cycle of rumination [20, 21]”: The two cited references do not provide support for the concept of "job embeddedness," as they do not investigate this construct. To support this claim, please cite literature that specifically defines and examines job embeddedness.

Response: References have been verified and updated.

6. Lines 234-247 ►“Trainee nurses […] job embeddedness process [22]”: This 15-line discussion passage contains overinterpretation, distortion of facts, or speculation presented as evidence.

Response: The relevant expressions have been revised.

7. Overall, the discussion section lacks synthesis and scholarly depth. I strongly advise you to review the discussion section in question and ensure every sentence that engages with a source can be directly and justifiably traced back to that source’s actual content.

Response: Thank you for your review. The discussion content has been revised.

VII. References

1. “Meihua et al.” is cited in the text but absent from the reference list.

2. “Crossley et al. ” is formatted incorrectly in the text per the journal style.

3. What does the symbol [J] at the end of the titles mean?

4. It would be wise to include the DOI when possible.

5. Please verify the title of the reference [3].

6. The entries for these references are incomplete (eg. [11], [14], [17]).

7. It appears that reference [18] is entirely fictitious.

Response: Thank you for your meticulous review. I have made the necessary revisions as requested.

Reviewer #2: The revised manuscript entitled “The interaction between rumination and job embeddedness among trainee nurses during internship: a multicenter longitudinal study” is generally well structured and clearly written. The introduction is informative; however, the authors should carefully verify that the references cited are fully consistent with the statements made. The methodology is clearly described, although the sampling method should be explicitly stated. The results are well presented and appropriately discussed. In addition, the limitations section should be expanded to include aspects beyond the small sample size. Finally, the authors are encouraged to carefully check and standardize all references according to the Vancouver referencing style.

Detailed comments and specific remarks are provided in the attached file.

Response: Thank you for your meticulous review. I have made the necessary revisions as requested.

Reviewer #3: O manuscrito é tecnicamente sólido por apresentar um delineamento longitudinal adequado, com a coleta de dados realizada em três momentos, com uso de abordagens estatísticas robustas e confiáveis (Modelo de defazagem cruzada e modelo paralelo de crescimento latente). A descrição dos métodos foi realizada de maneira clara e apropriada para as respostas as questões da pesquisa. As análises estatísticas forão rigorosas e apresentadas de forma bem transparente. Os indices de ajuste dos modelos utilizados forão reportados adequadamente, e o teste de viés reforçou a confiabilidade dos resultados.

É uma pesquisa original e de relevancia por analisar a ruminação e o enraizamento no trabalho em enfermeiros estagiários, devido principalmente ao contexto de formação clínica.

As conclusões vão de encontro aos dados apresentados , todas as afirmações são sustentadas. O manuscrito é claro, bem estruturado e intelegível para todos, as tabelas são pertinentes e contribuem para compreensão dos achados.

Foi apresentada a aprovação do Comitê de Ética e Pesquisa, e o estudo

---

## [Decision Letter · Decision Letter 1]

26 Feb 2026

PONE-D-25-60904R1The interaction between rumination and job embeddedness among trainee nurses duringinternship: a multicenter longitudinal studyPLOS One

Dear Dr. Ying,

Thank you for submitting your manuscript to PLOS ONE. After careful consideration, we feel that it has merit but does not fully meet PLOS ONE’s publication criteria as it currently stands. Therefore, we invite you to submit a revised version of the manuscript that addresses the points raised during the review process.

We look forward to receiving your revised manuscript.

Kind regards,

Ahmed Abdelwahab Ibrahim El-Sayed

Academic Editor

PLOS One

**Journal Requirements:**

**Additional Editor Comments:**

Dear Authors,

Thank you for your revision. We appreciate the improvements made to the manuscript. However, in this round of review, the reviewer has raised some minor comments that need to be addressed carefully before we can proceed to a final decision regarding your study.

Please revise the manuscript accordingly and provide a detailed response to the reviewer’s comments.

Reviewers' comments:

Reviewer's Responses to Questions

**Comments to the Author**

1. If the authors have adequately addressed your comments raised in a previous round of review and you feel that this manuscript is now acceptable for publication, you may indicate that here to bypass the “Comments to the Author” section, enter your conflict of interest statement in the “Confidential to Editor” section, and submit your "Accept" recommendation.

Reviewer #1: All comments have been addressed

Reviewer #2: (No Response)

Reviewer #3: All comments have been addressed

Reviewer #4: All comments have been addressed

2. Is the manuscript technically sound, and do the data support the conclusions?

Reviewer #1: Yes

Reviewer #2: Partly

Reviewer #3: Yes

Reviewer #4: Yes

3. Has the statistical analysis been performed appropriately and rigorously? 

Reviewer #1: Yes

Reviewer #2: I Don't Know

Reviewer #3: Yes

Reviewer #4: Yes

4. Have the authors made all data underlying the findings in their manuscript fully available?

Reviewer #1: Yes

Reviewer #2: Yes

Reviewer #3: Yes

Reviewer #4: No

5. Is the manuscript presented in an intelligible fashion and written in standard English?

Reviewer #1: Yes

Reviewer #2: Yes

Reviewer #3: Yes

Reviewer #4: Yes

6. Review Comments to the Author

Reviewer #1: (No Response)

Reviewer #2: The following comments are provided to clarify remaining issues that were not fully addressed in the revised manuscript and that require further attention to ensure scientific accuracy and transparency.

1. The reviewer’s previous comment has not been addressed.

The abstract still reports a recovery rate of 93.53% for 405 questionnaires, whereas the Results section clearly indicates that 382 valid questionnaires were collected, corresponding to an effective response rate of 94.32%.

Please correct the percentage reported in the abstract or provide a clear explanation for the discrepancy.

2. The revised manuscript does not adequately address the previous comment.

The statement claiming that “the correlation between persistent rumination and anxiety or depression is 2.5 times higher in trainee nurses than in the general population” remains unsupported by the cited literature.

The newly introduced reference does not report (i) data on trainee nurses, (ii) comparisons with the general population, nor (iii) effect size ratios indicating a 2.3–2.5-fold difference.

As currently written, this claim constitutes an unsubstantiated quantitative assertion and raises concerns regarding scientific accuracy.

Please either provide a source that explicitly reports this population-specific effect size or revise the statement to accurately reflect the available evidence.

3. The reviewer’s previous concern has not been addressed.

The revised manuscript still attributes specific percentage differences (19 % decrease in clinical performance accuracy and 41 % lower professional identity score) to Zeng et al. (2024).

However, this study is cross-sectional, does not include a control group, and does not report outcomes related to clinical performance accuracy or professional identity, nor any percentage-based group comparisons.

Merely changing the reference number does not resolve the issue. Please cite the correct source that explicitly reports these figures or remove/revise the statement accordingly.

4. The authors’ revision does not adequately address the reviewer’s concern.

4.1 Unsupported causal and intervention-related claims

Although the cited reference has been removed, the statement itself remains unchanged. As currently written, the paragraph presents causal and intervention-related claims (e.g., mitigation of negative emotions, enhancement of job embeddedness through psychological interventions) without empirical support.

Removing the citation without revising or qualifying the claim does not resolve the issue. The authors should either provide appropriate evidence supporting these assertions or clearly reformulate the text as a hypothesis or conceptual perspective rather than an evidence-based conclusion.

4.2 Continued inappropriate use of the same reference elsewhere in the manuscript

In addition, it should be noted that the previously removed reference (Geisler et al., 2023) is reused elsewhere in the manuscript in a manner that remains inappropriate. While Geisler et al. (2023) examine affective work rumination in relation to job demands and exhaustion, the study does not assess cognitive resource competition, professional role cognition, or organizational commitment.

These constructs are neither measured nor empirically discussed in the cited article. Therefore, interpretations attributed to this reference should be revised to clearly distinguish theoretical reasoning from empirically supported findings, or an appropriate reference should be provided.

5. The reviewer’s previous comment has not been addressed.

Although the sample size calculation and inclusion/exclusion criteria are clearly described, the manuscript still does not specify the sampling method or recruitment strategy.

It remains unclear whether participants were recruited via convenience sampling, cluster sampling by institution, or another approach, and how the 12 participating institutions and trainee nurses were selected.

Please explicitly state the sampling strategy and recruitment procedure to allow readers to assess potential selection bias and external validity.

6. The revision only partially addresses the reviewer’s comment.

While the statement regarding informed consent has been appropriately removed and the ethics approval ID has been correctly relocated as requested, the “Conflict of Interest” section still includes funding information.

Please separate the funding statement under a dedicated “Funding” subheading and retain only the conflict of interest declaration under “Conflict of Interest,” in line with standard journal structure. Further revision is required to fully address this point.

Reviewer #3: (No Response)

Reviewer #4: This manuscript does not contribute anything new to nursing science from a professional, academic, clinical, or other perspective. However, it is well-written, well-structured, easy to understand, and methodologically reproducible, except for one important flaw: the authors do not provide or make available the study's main component. This component is the self-developed electronic questionnaire distributed through the WenJuanxing platform (survey ID: 23045560). Furthermore, they do not provide the two main scales (RRS and GJEI) that the authors claim are included in the questionnaire. They also fail to state whether this questionnaire has been previously validated, which would be important and would strengthen the study. Therefore, my review focuses on the contribution of the questionnaires to ensure reproducibility, as this would be the study's greatest contribution. The results obtained have already been widely published in many fields and areas.

7. PLOS authors have the option to publish the peer review history of their article (what does this mean?). If published, this will include your full peer review and any attached files.

Reviewer #1: No

Reviewer #2: No

Reviewer #3: No

Reviewer #4: **Yes:**Ana Fernández-Araque

---

## [Author Response · Author response to Decision Letter 2]

9 Apr 2026

View Letter

Date: Feb 26 2026 08:27PM

To: "Chen Ying" 13814021690@163.com

From: "PLOS ONE" plosone@plos.org

Subject: PLOS ONE Decision: Revision required [PONE-D-25-60904R1]

PONE-D-25-60904R1

The interaction between rumination and job embeddedness among trainee nurses duringinternship: a multicenter longitudinal study

PLOS One

Dear Dr. Ying,

Thank you for submitting your manuscript to PLOS ONE. After careful consideration, we feel that it has merit but does not fully meet PLOS ONE’s publication criteria as it currently stands. Therefore, we invite you to submit a revised version of the manuscript that addresses the points raised during the review process.

A letter that responds to each point raised by the academic editor and reviewer(s). You should upload this letter as a separate file labeled 'Response to Reviewers'.

We look forward to receiving your revised manuscript.

Kind regards,

Ahmed Abdelwahab Ibrahim El-Sayed

Academic Editor

PLOS One

Journal Requirements:

Additional Editor Comments:

Dear Authors,

Thank you for your revision. We appreciate the improvements made to the manuscript. However, in this round of review, the reviewer has raised some minor comments that need to be addressed carefully before we can proceed to a final decision regarding your study.

Please revise the manuscript accordingly and provide a detailed response to the reviewer’s comments.

Reviewers' comments:

Reviewer's Responses to Questions

Comments to the Author

1. If the authors have adequately addressed your comments raised in a previous round of review and you feel that this manuscript is now acceptable for publication, you may indicate that here to bypass the “Comments to the Author” section, enter your conflict of interest statement in the “Confidential to Editor” section, and submit your "Accept" recommendation.

Reviewer #1: All comments have been addressed

Reviewer #2: (No Response)

Reviewer #3: All comments have been addressed

Reviewer #4: All comments have been addressed

2. Is the manuscript technically sound, and do the data support the conclusions?

Reviewer #1: Yes

Reviewer #2: Partly

Reviewer #3: Yes

Reviewer #4: Yes

3. Has the statistical analysis been performed appropriately and rigorously?

Reviewer #1: Yes

Reviewer #2: I Don't Know

Reviewer #3: Yes

Reviewer #4: Yes

4. Have the authors made all data underlying the findings in their manuscript fully available?

Reviewer #1: Yes

Reviewer #2: Yes

Reviewer #3: Yes

Reviewer #4: No

5. Is the manuscript presented in an intelligible fashion and written in standard English?

Reviewer #1: Yes

Reviewer #2: Yes

Reviewer #3: Yes

Reviewer #4: Yes

6. Review Comments to the Author

Reviewer #1: (No Response)

Reviewer #2: The following comments are provided to clarify remaining issues that were not fully addressed in the revised manuscript and that require further attention to ensure scientific accuracy and transparency.

1. The reviewer’s previous comment has not been addressed.

The abstract still reports a recovery rate of 93.53% for 405 questionnaires, whereas the Results section clearly indicates that 382 valid questionnaires were collected, corresponding to an effective response rate of 94.32%.

Please correct the percentage reported in the abstract or provide a clear explanation for the discrepancy.

Response: We appreciate the reviewer's thorough comments. Corresponding revisions have been made in the manuscript.

2. The revised manuscript does not adequately address the previous comment.

The statement claiming that “the correlation between persistent rumination and anxiety or depression is 2.5 times higher in trainee nurses than in the general population” remains unsupported by the cited literature.

The newly introduced reference does not report (i) data on trainee nurses, (ii) comparisons with the general population, nor (iii) effect size ratios indicating a 2.3–2.5-fold difference.

As currently written, this claim constitutes an unsubstantiated quantitative assertion and raises concerns regarding scientific accuracy.

Please either provide a source that explicitly reports this population-specific effect size or revise the statement to accurately reflect the available evidence.

Response: We appreciate the reviewer's thorough comments. The corresponding expressions and references have been modified in the manuscript.

3. The reviewer’s previous concern has not been addressed.

The revised manuscript still attributes specific percentage differences (19 % decrease in clinical performance accuracy and 41 % lower professional identity score) to Zeng et al. (2024).

However, this study is cross-sectional, does not include a control group, and does not report outcomes related to clinical performance accuracy or professional identity, nor any percentage-based group comparisons.

Merely changing the reference number does not resolve the issue. Please cite the correct source that explicitly reports these figures or remove/revise the statement accordingly.

Response: We appreciate the reviewer's thorough comments. The relevant expressions have been modified in the manuscript.

4. The authors’ revision does not adequately address the reviewer’s concern.

4.1 Unsupported causal and intervention-related claims

Although the cited reference has been removed, the statement itself remains unchanged. As currently written, the paragraph presents causal and intervention-related claims (e.g., mitigation of negative emotions, enhancement of job embeddedness through psychological interventions) without empirical support.

Removing the citation without revising or qualifying the claim does not resolve the issue. The authors should either provide appropriate evidence supporting these assertions or clearly reformulate the text as a hypothesis or conceptual perspective rather than an evidence-based conclusion.

Response: We appreciate the reviewer's thorough comments. The relevant expressions have been modified in the manuscript.

4.2 Continued inappropriate use of the same reference elsewhere in the manuscript

In addition, it should be noted that the previously removed reference (Geisler et al., 2023) is reused elsewhere in the manuscript in a manner that remains inappropriate. While Geisler et al. (2023) examine affective work rumination in relation to job demands and exhaustion, the study does not assess cognitive resource competition, professional role cognition, or organizational commitment.

These constructs are neither measured nor empirically discussed in the cited article. Therefore, interpretations attributed to this reference should be revised to clearly distinguish theoretical reasoning from empirically supported findings, or an appropriate reference should be provided.

Response: We appreciate the reviewer's suggestion. The text has been modified to reflect a hypothetical tone.

5. The reviewer’s previous comment has not been addressed.

Although the sample size calculation and inclusion/exclusion criteria are clearly described, the manuscript still does not specify the sampling method or recruitment strategy.

It remains unclear whether participants were recruited via convenience sampling, cluster sampling by institution, or another approach, and how the 12 participating institutions and trainee nurses were selected.

Please explicitly state the sampling strategy and recruitment procedure to allow readers to assess potential selection bias and external validity.

Response: We appreciate the reviewer's thorough comments. The relevant expressions have been modified in the manuscript.

6. The revision only partially addresses the reviewer’s comment.

While the statement regarding informed consent has been appropriately removed and the ethics approval ID has been correctly relocated as requested, the “Conflict of Interest” section still includes funding information.

Please separate the funding statement under a dedicated “Funding” subheading and retain only the conflict of interest declaration under “Conflict of Interest,” in line with standard journal structure. Further revision is required to fully address this point.

Response: We appreciate the reviewer's thorough comments. The relevant expressions have been modified in the manuscript.

Reviewer #3: (No Response)

Reviewer #4: This manuscript does not contribute anything new to nursing science from a professional, academic, clinical, or other perspective. However, it is well-written, well-structured, easy to understand, and methodologically reproducible, except for one important flaw: the authors do not provide or make available the study's main component. This component is the self-developed electronic questionnaire distributed through the WenJuanxing platform (survey ID: 23045560). Furthermore, they do not provide the two main scales (RRS and GJEI) that the authors claim are included in the questionnaire. They also fail to state whether this questionnaire has been previously validated, which would be important and would strengthen the study. Therefore, my review focuses on the contribution of the questionnaires to ensure reproducibility, as this would be the study's greatest contribution. The results obtained have already been widely published in many fields and areas.

Response: We appreciate the reviewer's suggestion. The relevant scales, along with validation data on their reliability and validity in China, have been added to the Methods section.

7. PLOS authors have the option to publish the peer review history of their article (what does this mean?). If published, this will include your full peer review and any attached files.

Do you want your identity to be public for this peer review? For information about this choice, including consent withdrawal, please see our Privacy Policy.

Reviewer #1: No

Reviewer #2: No

Reviewer #3: No

Reviewer #4: Yes: Ana Fernández-Araque

---

## [Decision Letter · Decision Letter 2]

30 Apr 2026

The interaction between rumination and job embeddedness among trainee nurses duringinternship: a multicenter longitudinal study

PONE-D-25-60904R2

Dear Authors,

We’re pleased to inform you that your manuscript has been judged scientifically suitable for publication and will be formally accepted for publication once it meets all outstanding technical requirements.

Kind regards,

Ahmed Abdelwahab Ibrahim El-Sayed,

Academic Editor

PLOS One

Additional Editor Comments

Dear Authors,

Thank you for submitting the revised version of your manuscript and for your careful attention to the reviewers’ comments. The revisions have significantly improved the quality and clarity of the manuscript.

I am pleased to inform you that your manuscript has now been accepted for publication.

Congratulations, and we look forward to seeing your work published.

Reviewers' comments:

Reviewer's Responses to Questions

**Comments to the Author**

1. If the authors have adequately addressed your comments raised in a previous round of review and you feel that this manuscript is now acceptable for publication, you may indicate that here to bypass the “Comments to the Author” section, enter your conflict of interest statement in the “Confidential to Editor” section, and submit your "Accept" recommendation.

Reviewer #2: All comments have been addressed

2. Is the manuscript technically sound, and do the data support the conclusions?

Reviewer #2: Yes

3. Has the statistical analysis been performed appropriately and rigorously? 

Reviewer #2: I Don't Know

4. Have the authors made all data underlying the findings in their manuscript fully available?

Reviewer #2: Yes

5. Is the manuscript presented in an intelligible fashion and written in standard English?

Reviewer #2: Yes

6. Review Comments to the Author

Reviewer #2: The authors have addressed all the comments raised during the previous rounds of review. The revisions have helped correct the identified inconsistencies and improve the scientific accuracy of the manuscript. In particular, previously unsupported claims have been revised or reformulated more appropriately, and the overall presentation of the text has gained in clarity and coherence. Methodological aspects, especially those related to the description of the sampling strategy and the manuscript structure, have also been improved in line with the recommendations. In its current version, the manuscript no longer presents major issues that would compromise its scientific validity. Therefore, I consider that the manuscript can be accepted for publication in its revised form.

7. PLOS authors have the option to publish the peer review history of their article (what does this mean?). If published, this will include your full peer review and any attached files.

Reviewer #2: **Yes:**Dr. Abdelmonim El Fadely

---

## [Editor Report · Acceptance letter]

PONE-D-25-60904R2

PLOS One

Dear Dr. Ying,

I'm pleased to inform you that your manuscript has been deemed suitable for publication in PLOS One. Congratulations! Your manuscript is now being handed over to our production team.

Kind regards,

on behalf of

Dr. Ahmed Abdelwahab Ibrahim El-Sayed

Academic Editor

PLOS One